# A pilot acceptability evaluation of MomMind: A digital health intervention for Peripartum Depression prevention and management focused on health disparities

**Alexandra Zingg** *, **Amy Franklin, Angela Ross, Sahiti Myneni**

Department of Clinical and Health Informatics, McWilliams School of Biomedical Informatics at the University of Texas Health Science Center Houston, Houston, Texas, United States of America

* azingg2@gmail.com

**Data Availability Statement:** Data leading to this study findings is available without restriction within the manuscript and in our supplementary files.

## Abstract

Health disparities cause significant strain on the wellbeing of individuals and society. In this study, we focus on the health disparities present in the condition of Peripartum Depression (PPD), a significant public health issue. While PPD can be managed through therapy and medication, many women do not receive adequate PPD treatment due to issues of social stigma and limited access to healthcare resources. Digital health technologies can offer practical tools for PPD management. However, current solutions do not integrate behavior theory and are rarely responsive to the transient information needs stemming from women's unique sociodemographic, clinical and psychosocial profiles. We describe a pilot acceptability evaluation of MomMind, a health-disparities focused digital health intervention for the prevention and management of PPD. A crucial MomMind advantage is its basis on behavior change theory and patient engagement as enabled by the Digilego digital health framework. Following an internal usability evaluation, MomMind was evaluated by patients through cross-sectional acceptability surveys, pre-and-post PPD health literacy surveys, and interviews. Survey respondents included n = 30 peripartum women, of whom n = 16 (53.3%) were Hispanic and n = 17 (56.7%) of low-income. Survey results show that 96.6% of participants (n = 29) approved and welcomed MomMind, and 90% (n = 27) found MomMind to be an appealing intervention. Additionally, significant improvements (p< = 0.05) were observed in participants' PPD health literacy, specifically their ability to recognize PPD symptoms and knowledge of how to seek PPD information. Interview main themes include MomMind's straightforward design and influence of others (family members, providers) on use of technology. Results suggest that enhancement of a digital health framework with health literacy theory can support production of digital health solutions acceptable to vulnerable populations. This study incorporates existing theories from different disciplines into a unified approach for mitigating health disparities, and produced a novel solution for promotion of health in a vulnerable population.

**Funding:** Research reported in this publication was partly supported by the National Library of Medicine of the National Institutes of Health (www. nlm.nih.gov) under award numbers 1R01LM012974-01A1 (SM). The funders had no role in study design, data collection and analysis, decision to publish, or preparation of the manuscript.

**Competing interests:** The authors have declared that no competing interests exist.

## Author summary

In this work, patients were given a first look to a website under construction and containing: a) patient education on depression during pregnancy and shortly after birth, b) mood and depression symptom trackers, c) a social network section specifically to discuss depression in new/expectant moms, and d) a diary for open writing. The purpose of this work was to learn what patients liked and disliked about the website, and to document their first impressions. Before showing them the website, we documented what patients already knew, believed, and thought about depression during pregnancy and shortly after birth. We noticed that immediately after showing them the website, patients felt much more confident in their ability to look for trustworthy information about depression in pregnancy and after birth. Considering that most of our patients were Hispanic and of limited economic resources, this finding gives us confidence that our materials and features can empower women who need it the most to seek needed information for their mental well-being during a challenging time in their lives. Patients enjoyed the design of the website and what they learned from it, and they indicated that they would share it with their friends, family members, and other women.

## Introduction

Health disparities cause significant strain on the physical, psychological, and financial wellness of individuals and the healthcare system as a whole [1]. Advances in digital health technologies offer a potential pathway to help reduce or eliminate health disparities through improved access to care and patient education [2]. However, previous studies have indicated failures of digital health technologies in providing optimal healthcare information and resources to vulnerable populations [3,4]. Understanding the particular information and technology needs of these populations is critical to achieve better and more equitable digital health solutions.

Here, we focus on the health disparities present in the condition of Peripartum Depression (PPD), a significant public health issue that affects approximately one in seven peripartum women [5]. Depression in women is associated with significant comorbidities and burden of disease, and previous research indicates that PPD and other peripartum mood disorders incur financial costs of approximately $31,800 per each mother-infant dyad [6]. Rates of PPD are higher among women of low socioeconomic status, minorities, and immigrants [7,8]. While PPD can be managed through therapy and medication, many women do not receive adequate PPD treatment due to issues of social stigma, low access to mental health resources, and not having available childcare or time off work [9]. Untreated PPD may lead to adverse health outcomes such as low birthweight, and infants of women with PPD are more likely to exhibit behavioral and cognitive issues [10].

In this study, we describe a pilot evaluation of MomMind, a health-disparities focused digital health intervention for the prevention and management of PPD. In Table 1, we describe the basic features of the intervention. These features were designed and developed based on our previous user needs assessment and theoretical mapping [11–15].

We conduct the prototype evaluation with a sample of majority low-socioeconomic status (SES) perinatal women seeking care at a high-risk pregnancy clinic. The purpose of the evaluation is to capture intended users' perceived acceptability, feasibility, and appropriateness of the tool. We also assess changes in participants' PPD health literacy through measuring their knowledge, attitudes, and beliefs about PPD before and after use of the tool. Furthermore, we qualitatively gather their initial impressions about the tool with individual interviews. The

**Table 1. MomMind Features.**

| MomMind Feature | Description |
| --- | --- |
| "My Diary" | Digital journal where the user can document thoughts |
| "Mom Talk" | Social forum where participants can ask questions about PPD |
| "My Library" | Library repository of PPD patient education materials (the "PPD 101" series). A complete documentation of scripts and sources used for the "PPD 101" video series can be found in S1 Appendix. Videos can also be accessed from https://youtube.com/playlist?list=PLUlN-0kA-J1uu7spHqDIQReZUrSzAC39- |
| "My Surveys" | Repository of PPD self-monitoring surveys |

main aim of this study is to address a research question that has been seldom addressed in the literature: Will a digital health tool that was designed and developed through a specialized framework incorporating sociobehavioral, health literacy, and technological theories be acceptable among a representative sample of a vulnerable population? We expect that through the use of such a specialized digital health framework we will produce a digital health solution for PPD management that will have high levels of acceptability among a sample of low-income perinatal women. Our study, in turn, can help guide the future production of digital health tools for health promotion.

## Methods

### Study design

Our mixed-methods study is represented in Fig 1, where we illustrate our comprehensive MomMind design and development process from user needs analysis to evaluation.

### Acceptability evaluation

Our acceptability evaluation of MomMind was conducted following an internal heuristic evaluation at the McWilliams School of Biomedical Informatics Center for Digital Health and Analytics. Our acceptability evaluation consists of cross-sectional and pre-post surveys, as well as one-on-one interviews. For our study sample, a total of 31 low-SES perinatal women were enrolled in our study. This study sample was recruited over a period of approximately two months. Our study sample size is in accordance with statistical recommendations indicating that sample sizes between approximately n = 26 to n = 34 are satisfactory for studies gauging initial acceptability/feasibility of a process or product, as the emphasis of such studies is to capture initial trends, preliminary insights, and qualitative feedback rather than statistical significance [16].

**Inclusion criteria.** Participants had to be English speaking, at least 18 years old, and be a current patient at the UT Physicians Fetal Center located in the Texas Medical Center. The clinic treats high-risk pregnancy conditions such as multiple births and placental disorders. Based on a previous study [13], we expected most participants to qualify as low-income based on U.S. Department of Housing and Urban Development guidelines [17]. Patients attending the clinic for their regular prenatal or postnatal care were invited to participate in the study through referrals from clinical collaborators (their OBGYNs, clinic Nursing staff) and were offered a $25 gift card as compensation for participating in the study. Patients who agreed to participate provided informed consent. Approval for this study was given by the UT Health Committee for Protection of Human Subjects IRB HSC-SBMI-22-0750.

**Evaluation procedures.** Once participants provided informed consent, a sociodemographic survey was administered to collect information on characteristics such as ethnicity,

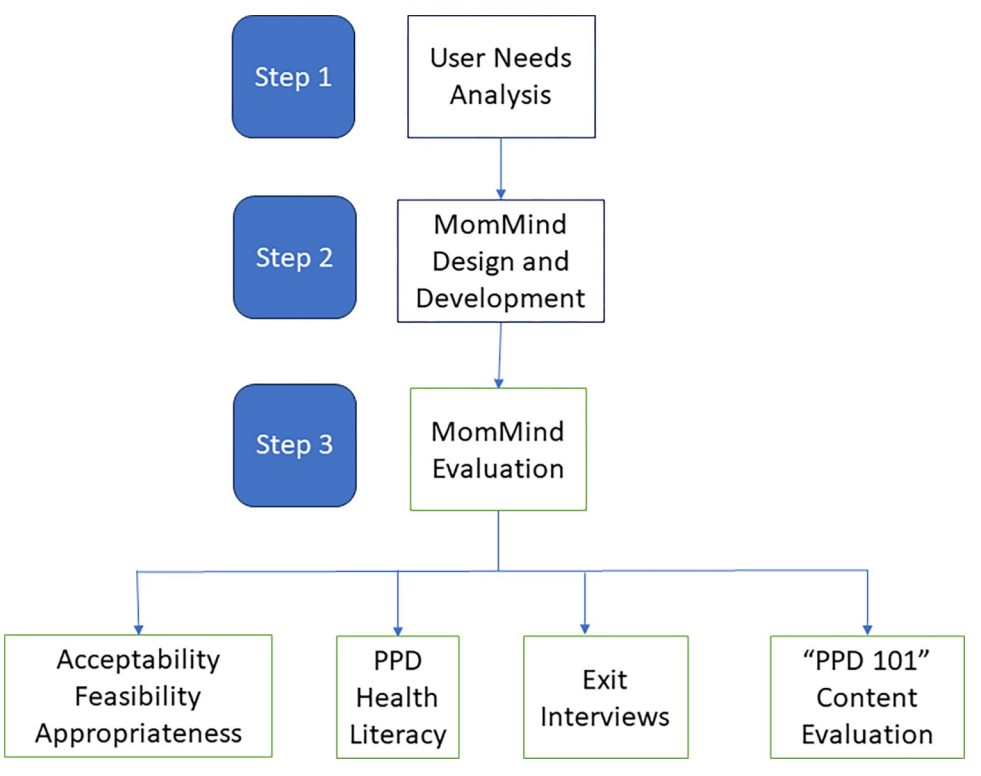

**Fig 1. Methodological Overview.**

income, and education (S2 Appendix). Participants were introduced to MomMind through a walkthrough and description of the tool, and asked to complete the activities described in Table 2.

All data collected from participants during this study was securely stored behind protected university servers. Participants were informed that all study responses are solely for research purposes and kept private and confidential.

**Capturing MomMind acceptability, feasibility, and appropriateness.** After participants completed all tasks, they responded to Weiner's survey battery consisting of three brief

**Table 2. Evaluation Procedures.**

| Task | Description |
|---|---|
| Writing an online forum post in "MomTalk" | Write a post similar to ones in popular online forums (i.e., Facebook, Instagram), about any topic of choice. |
| Writing an entry in "My Diary" | Use the tool as you would any diary and write about topics such as feelings, reminders, questions, and pregnancy milestones. |
| Going into the "My Surveys" feature and selecting their present mood with the Pick-a-Mood scale [18] | The Pick-a-Mood scale presents to the user a wheel of picture representations of different moods (i.e. relaxed, excited, neutral) that the user can select. The user can only select and submit one mood at a time (S2 Appendix). |
| Completing a printed version of the Edinburgh Postnatal Depression Scale (EPDS, S2 Appendix) [19] | The EPDS contains ten items that assess peripartum women's mental wellbeing and the presence of depression symptoms. A score of 10 or above (of a possible maximum of 30) in the EPDS is indicative of possible depression. |
| Receiving patient education on PPD | Selecting at least one library module in "My Library" and watching at least one "PPD 101" video. |

measures: 1) Acceptability of Intervention Measure, 2) Feasibility of Intervention Measure, and 3) Intervention Appropriateness Measure [20]. These measures contain four Likert-scale items each (S3 Appendix). All three measures have excellent internal consistency, with the Cronbach's alpha for the Acceptability of Intervention scale being $\alpha = 0.85$, for Feasibility $\alpha = 0.89$, and for Appropriateness $\alpha = 0.91$.

**Capturing Peripartum Depression health literacy.** Both before and immediately after completing the MomMind tasks, participants answered the Postpartum Depression Literacy Scale (PoDLiS) [21]. The PoDLiS instrument uses a 5-point likert scale system (1 = strongly disagree or not likely at all and 5 = strongly agree or very likely) to assess user's PPD literacy levels for the following seven attributes: 1) ability to recognize PPD, 2) attitudes which facilitate recognition of PPD and appropriate help seeking, 3) knowledge and beliefs of self-care activities, 4) knowledge of how to seek information related to PPD, 5) beliefs about professional help available, 6) knowledge about professional help available, and 7) knowledge of PPD risk factors and causes (S2 Appendix). In the PoDLiS scale, a high score corresponds to high PPD literacy levels, while a low score indicates low levels of PPD literacy. The exceptions to this are for two of the literacy attributes (attitudes which facilitate recognition of PPD and appropriate help seeking, and beliefs about professional help available), where a lower score corresponds to higher literacy levels. The PoDLiS scale has shown significant internal consistency and has a Cronbach's alpha coefficient of $\alpha = 0.78$.

**Individual interviews.** A one-on-one exit interview was conducted where participants verbally shared their personal opinions regarding MomMind. Our interview questions are based on the Integrated Behavioral Model, a widely used model in the discipline of public health [22]. In this model, the focus is on an individual's intent to perform a specific health behavior. The model contains five specific constructs that are described below, along with corresponding interview questions used during MomMind exit interviews:

a.  Experiential attitude: an indvidual's personal experience when performing a behavior. Interview questions: How do you feel about the idea of using MomMind? What did you like the most about MomMind? What did you like the least?)

b.  Instrumental Attitude: an individual's expected results from performing a behavior. Interview questions: What are the benefits that might result from using MomMind? What are the negative effects that might result from using MomMind?

c.  Normative Influence: perceived influence of others in performing a behavior. Interview questions: Who would support your using MomMind?, Who would be against your using MomMind?

d.  Perceived Control: an individual's perceived level of control in performing a behavior. Interview questions: What things make it easy for you to use MomMind?, What things make it hard for you to use MomMind?

e.  Self-Efficacy: an individual's level of confidence in their ability to perform a behavior. Interview question: If you want to use MomMind, how certain are you that you can?

**"PPD 101" content evaluation.** After the exit interview, participants were asked to complete a feedback survey for the educational content of MomMind. Survey items were adapted from prior research related to digital interventions and health content evaluation [23]. Our survey contains five Likert-scale items (1 = Strongly Disagree, 5 = Strongly Agree): 1) The PPD 101 videos helped improve my basic understanding of peripartum depression, 2)Would you recommend the PPD 101 videos to friends and family?, 3)The PPD 101 videos were easy to

understand, 4) I enjoyed the graphics and audio of the PPD 101 videos, and 5) I would want to watch other PPD 101 videos to learn more.

### Data analysis

Descriptive statistical measures of proportions were calculated for participant demographics and all acceptability survey results. For analysis of PPD health literacy scores, statistical differences between pre-and-post intervention scores were analyzed using the Wilcoxon-Signed Rank Test [24]. The use of the Wilcoxon-Signed Rank Test is appropriate, as Likert-scale data is not normally distributed [25]. Additionally, a sub-analysis of PPD health literacy scores was done for participants living in low-income households. For the qualitative data of interviews, transcripts were entered into the Dedoose qualitative analysis software and analyzed using grounded theory [26]. Through this approach, a line-by-line analysis of the data is first done to extract individual ideas and concepts (also called open codes) and then major themes are derived from patterns and connections in the open codes. In this study, a single researcher independently coded all transcripts, and a second researcher also coded a subset of five transcripts to ensure coding objectivity. Code comparison was constantly used to ensure code agreement and consistency among coders, and from this process a final list of core MomMind acceptability themes was derived.

## Results

### Acceptability evaluation

**Participant characteristics.** Of 31 perinatal women who enrolled in the study, 29 completed the demographic survey. In total, 17 out of 30 participants resided in households of low-income ranges. A total of 18 participants had a level of education of less than a college degree. Participant demographics are summarized in Table 3.

**Table 3. Participant Demographics.**

| Ethnicity | N (%) |
| --- | --- |
| Hispanic | 16 (55%) |
| Non-Hispanic White | 9 (31%) |
| Black | 3 (10%) |
| Asian | 1 (4%) |
| **Household Income** | **N (%)** |
| <$25,000 | 5 (17%) |
| $25,000-$39,999 | 6 (21%) |
| $40,000-$69,999 | 6 (21%) |
| $70,000-$99,999 | 3 (10%) |
| $100,000- $149,999 | 3 (10%) |
| $150,000-$199,999 | 5 (17%) |
| <$200,000 | 1 (4%) |
| **Education** | **N (%)** |
| Some high school | 1 (3%) |
| High school | 6 (21%) |
| Technical/Vocational Training | 4 (14%) |
| Some college credit | 6 (21%) |
| Associate's Degree | 1 (3%) |
| Bachelors' Degree | 8 (28%) |
| Doctoral Degree | 3 (10%) |

**Peripartum Depression symptoms.**   According to responses from the EPDS survey, participants had an average EPDS score of 6.6 (out of 30). The minimum score was 0, maximum score was 19, and standard deviation was 4.45. In total, 8 participants met the EPDS cut-off score of 10 or higher for possible depression.

**MomMind acceptability.**   Of the 31 patients who enrolled for the MomMind evaluation, 30 completed Weiner's survey battery for intervention acceptability, feasibility and appropriateness. Survey results show that 96.6% (n = 29) of respondents were above neutral (agreed or completely agreed) in approving of MomMind as a PPD prevention and management intervention; another 96.6% were above neutral in welcoming and liking MomMind, and 90% (n = 27) found MomMind to be an appealing intervention. Three respondents were neutral regarding the appeal of MomMind, and there was one neutral response each for the items of approval, liking, and welcoming of MomMind. For more details on the acceptability, feasibility, and appropriateness scores for MomMind, please see S3 Appendix.

**MomMind feasibility.**   Our results show that all respondents were above neutral in classifying MomMind as an intervention that was doable for them, while 96.6% were above neutral in deeming MomMind as easy to use and on the possibility of their using MomMind. A total of 90% (n = 27) of respondents considered MomMind an implementable intervention. Three respondents (10%) were neutral on MomMind being an implementable intervention, and one respondent (3%) was neutral on MomMind's ease of use. One respondent disagreed with the possibility of using MomMind.

**MomMind appropriateness.**   All respondents regarded MomMind as a suitable intervention for PPD management. A majority of 96.6% (n = 29) also regarded MomMind as an applicable and fitting intervention for them. A total of 93.3% (n = 28) answered that the MomMind intervention was a good match for them. Two respondents were neutral on MomMind being a good match for them. Survey items regarding the fitting and applicability of MomMind received one neutral response each.

**Peripartum Depression health literacy.**   A majority of our sample (70% or n = 21 out of 30 respondents) recognized unusual sadness and being tearful as possible symptoms of PPD and responded above neutral (Somewhat agree or strongly agree) to the corresponding item. The PPD aspect that went most unrecognized by participants was that symptoms last for a period of at least two weeks, with a majority (60% or n = 18) responding neutral (Neither agree nor disagree) or below neutral (Somewhat disagree or strongly disagree) to the corresponding item. Participants' average score (on a 1–5 Likert scale) for ability to recognize PPD symptoms was 3.89 at pre-intervention and 4.25 at post-intervention, an improvement of statistical significance ($p = 0.05$).

In terms of PPD risk factors and causes, the risk factor that participants most associated with PPD was stressful life circumstances (i.e., death of a loved one or divorce), with 90% responding above neutral (Extremely likely or somewhat likely). The risk factors that participants least associated with PPD was genetics or inherited problems, with 50% responding neutral (neither likely nor unlikely) or below (somewhat unlikely or extremely unlikely). Average score on knowledge of PPD risk factors and causes at pre-intervention was 4.23, and at post-intervention 4.27, a change that is statistically insignificant.

Out of various self-care activities (i.e., physical activity, a balanced diet) participants felt most strongly about seeking help for household chores and infant care as a preventer and manager of PPD, with 90% scoring above neutral. Average score on knowledge and beliefs of self-care activities at pre-intervention was 4.05, and at post-intervention 4.28, showing a statistically significant increase.

For knowledge of how to seek information about PPD, participants felt most confident regarding the use of various sources to seek information about PPD, with 86.6% (n = 26)

scoring above neutral for the item. Pre-intervention scores for knowledge of how to seek information related to PPD were on average 3.49, and at post-intervention 3.79 (a statistically significant improvement). For more details on all scores for PPD health literacy attributes, please see S3 Appendix.

**Peripartum Depression health literacy among low-income participants.**   Participants of low-income ranges showed a statistically significant improvement for the PPD health literacy attribute of knowledge of how to seek information related to PPD. For this attribute, participants had an average score of 3.31 at pre-intervention and 3.64 at post-intervention. No other statistically significant changes were observed for these participants' PPD health literacy scores.

**Interview themes.**   There were five main themes that emerged from individual interviews, ranging from the positive characteristics of feature design to recommendations for improvement. These themes are further described and illustrated with individuals' comments in Table 4.

**"PPD 101" content evaluation.**   Results of our "PPD 101" content evaluation showed that 90% of participants (n = 27) agreed the "PPD 101" videos were easy to understand. Another 83.34% (n = 25) of participants agreed that the video graphics and audio were enjoyable, and would want to watch more videos for more PPD information. And, 79% (n = 24) scored above neutral (agreed or strongly agreed) to recommending the videos to their friends and family (S3 Appendix).

## Discussion

This study aims to assess the level of acceptability a PPD digital health tool developed through a framework focused on health disparities will have among low-income perinatal women. We optimized MomMind for inclusivity through the participation of women from minority and low-income backgrounds in our user needs analysis. We also ensured that our "PPD 101" content is easy to process (such as ensuring levels of readability according to our users' education background) and representative of our target population. Our evaluation results show that MomMind had an excellent acceptance rate by a sample of majority low-income and Hispanic peripartum women. Our results are in line with existing studies on the impact of digital health technologies among similar populations; as one example, Bhat and colleagues (2018) [27] conducted a study evaluating a two-way text messaging feature as a complement to depression management and education among a sample of rural and low-income peripartum women living in the state of Washington. Results for the study showed that 94% of the women found the messaging feature to be helpful. Another noteworthy find from our study is the significant improvement in certain PPD health literacy attributes (ability to recognize PPD symptoms and knowledge on how to seek PPD information) for study participants. Through our evaluation, we were also able to identify some PPD knowledge gaps for our target population. For example, most participants were unaware that PPD symptoms typically last for a period of more than two weeks. We also observed that MomMind was effective in improving knowledge of how to seek PPD information among those participants of low income. This finding suggests that the use of a health disparities-focused digital health framework for the production of digital health content and features is indeed effective in addressing health literacy gaps present in underserved populations. Participants showed a significantly positive acceptance rate toward the "PPD 101" content presented to them, and they specifically valued its easy access to credible information resources and multimedia format. Furthermore, the "PPD 101" content sparked participants' curiosity in obtaining more knowledge about PPD and sharing the "PPD 101" videos with others. Allowing participants more interaction time with MomMind would have likely resulted in detection of further PPD health literacy improvements.

**Table 4. Interview Themes.**

| Theme | Participant Comments |
|---|---|
| Positive Characteristics of Feature Design: Individuals provided positive feedback on MomMind's straightforward structure and its ease of navigation. | P2: "It is really short and sweet, to the point . . . I do like how everything's real precise". <br> P21: "I really liked the nutrition module that I picked. I like how there's lists broken down, kind of like a question-answer format." <br> P28: "It seems pretty straightforward. I like that. It's not just like a whole page. I kept the accordion one at a time. . . less in your face if that information wasn't good for you." |
| Influencers: Individuals identified close family members, their medical team, and even themselves as encouragers and promoters of the MomMind app. | P11: "My husband has a different application for me being pregnant. I probably would be good on there [MomMind] and read some stuff, so he would probably be an encourager for sure." <br> P27: "Myself. You know, I have struggled with anxiety and depression in the past, and so I am very open to getting help and having resources available to me." <br> P28: "I think the nursing staff would be good representatives. Like, 'Hey I have this resource for you guys', as far as a patient and someone that works here, I think it'd be good. Maybe not that first visit like the initial phone call, but like, 'Hey, you're getting close to having your baby'." |
| Benefits of app for PPD prevention: Individuals commented on how each MomMind feature can provide help for PPD prevention and management. As example, benefits of the My Library module included its multimedia format and reliable information. | P1: "I think it'll help become more informed about postpartum depression and what to do because I remember my last one, I didn't get as much information as I did from here and it was a lot harder to get over it. . .I feel like this will help anybody who doesn't want to ask questions up front". <br> P2: "I liked the library the most, because it was visual and then it was something else I could read, kind of take with me". |
| Recommendations for Improvement: Participants recommended the addition of new features to MomMind content, such as video voice-over and editing of video fonts to better suit their on-the-go lifestyles. Features like calendars and alerts were also mentioned by participants. | P21: "It'd be nice if voice-over was made for the videos, because I could listen while working". <br> P30: "I would suggest a calendar that could send alerts." <br> P2: "I would need little pep talks. . . like, 'Your appointment is coming up, just checking in'". |
| Modern Aspects of App: Participants valued the app's focus on PPD and its applicability to a large population of women; they also stated how having a central system containing various digital features for PPD prevention and management was an innovative characteristic of the app. | P1: "I just know that this would probably help a lot of women" <br> P9: "It's something new to me, it's a lot different than Facebook." <br> P14: "It's good to have one application for everything: resources, talking to others, writing for yourself. . ." |

One of the most salient results of our individual interviews was how participants were able to identify various benefits of MomMind, such as being able to socialize with peers and the availability of credible PPD education materials. Participants also had positive remarks on MomMind having a simplistic design and being user-friendly. However, they also had recommendations for improving MomMind, such as the need for more informational feedback after finishing self-monitoring surveys and other digital features like calendars, notifications, and stratifying MomTalk participants by due date. All of these participant suggestions are to be considered for future versions of MomMind.

A possible limitation of our study is its cross-sectional component, where participants had limited interaction time with MomMind. It is likely that a longitudinal design could provide a more robust evaluation of user experience with MomMind. Nevertheless, our cross-sectional

design offers the advantage of reducing the potentially high participant attrition rates that are more probable in longitudinal studies [28]. Another study limitation was the exclusion of Spanish-speaking women, limiting the generalizability of study results. This group of women in particular may experience multiple barriers to PPD care, such as cultural norms prompting them to seek help mainly within their family [29], and poor quality of care due to language barriers and discrimination [30]. In the future, plans for MomMind include: the recruitment of Spanish-speaking participants as part of our user needs analysis, adaptation of MomMind content for various cultures and languages, and the pursual of further evaluation studies using high-rigor designs such as Randomized Controlled Trials. Longer term plans for MomMind include eventual implementation of the intervention in clinical settings relevant to PPD (such as Neonatal Intensive Care Unit, Psychiatry, OB/GYN). Additionally, we will pursue partnerships with researchers in developing countries to evaluate the performance of the tool in these settings.

## Conclusion

Future digital health interventions targeting vulnerable populations should consider the use of specialized digital health frameworks addressing social determinants of health such as health literacy (an example being the extended Digilego framework we have presented in this study) for intervention design and development. Previous studies that also presented digital health tools targeting vulnerable populations have used approaches such as participatory design [31] and community-based participatory research [3]. Other studies have also employed the eHealth literacy framework during development of digital interventions in areas ranging from nutrition and aging [32] to non-communicable diseases [33]. While these efforts represent an advancement towards digital health equity, there is still a significant scarcity of research that considers the underlying theories behind the development of these particular tools, a factor that can diminish the effectiveness of these tools and which can contribute to an existing digital health divide [34]. Our MomMind evaluation results show that our specialized digital health framework resulted in a product that is widely acceptable to our target population, and furthermore they suggest that our specialized content design and development strategy significantly improved our participants' PPD health literacy levels. The framework presented in this study can be applied to health domains outside of PPD, such as chronic diseases, by adapting our digital feature and content engineering based on corresponding user needs analysis and theory integration. With the increase of health disparities [1] and the rapidly changing demographics of populations in the U.S. and worldwide [35], the introduction of specialized theoretical frameworks for digital health design and development is timely and can help improve health equity and promotion among various populations and social settings.

## Supporting information

**S1 Appendix. "PPD 101" Scripts.**
(DOCX)

**S2 Appendix. Surveys and Questionnaires.**
(DOCX)

**S3 Appendix. MomMind Evaluation Results.**
(PPTX)

**S1 Data. MomMind Evaluation Data.**
(ZIP)

## Acknowledgments

We would like to thank all patients and clinicians who dedicated their time and effort to the completion of this study. The content of this study is solely the responsibility of the authors and does not necessarily represent the official views of the NIH.

## Author Contributions

**Conceptualization:** Alexandra Zingg, Amy Franklin, Angela Ross, Sahiti Myneni.

**Formal analysis:** Alexandra Zingg, Sahiti Myneni.

**Funding acquisition:** Sahiti Myneni.

**Investigation:** Alexandra Zingg, Sahiti Myneni.

**Methodology:** Alexandra Zingg, Amy Franklin, Angela Ross, Sahiti Myneni.

**Resources:** Sahiti Myneni.

**Software:** Alexandra Zingg.

**Supervision:** Amy Franklin, Angela Ross, Sahiti Myneni.

**Writing – original draft:** Alexandra Zingg.

**Writing – review & editing:** Amy Franklin, Angela Ross, Sahiti Myneni.

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
