## [Decision Letter · Decision Letter 0]

30 Nov 2023

PDIG-D-23-00368

A Pilot Acceptability Evaluation of MomMind: A Digital Health Intervention for Peripartum Depression Prevention and Management Focused on Health Disparities.

PLOS Digital Health

Dear Dr. Zingg,

Thank you for submitting your manuscript to PLOS Digital Health. After careful consideration, we feel that it has merit but does not fully meet PLOS Digital Health's publication criteria as it currently stands. Therefore, we invite you to submit a revised version of the manuscript that addresses the points raised during the review process.

Please submit your revised manuscript within 30 days Dec 30 2023 11:59PM. If you will need more time than this to complete your revisions, please reply to this message or contact the journal office at digitalhealth@plos.org. Please include the following items when submitting your revised manuscript:

We look forward to receiving your revised manuscript.

Kind regards,

Kara Burns

Guest Editor

PLOS Digital Health

Journal Requirements:

2. Please provide separate figure files in .tif or .eps format only and remove any figures embedded in your manuscript file. Please also ensure that all files are under our size limit of 10MB.

3. We do not publish any copyright or trademark symbols that usually accompany proprietary names, eg ©, ®, ™ (e.g. next to drug or reagent names). Please remove all instances of trademark/copyright symbols throughout the text, including © on page 20.

4. In the online submission form, you indicated that "Datasets generated and/or analyzed during the current study are not publicly available for protection of individual privacy, but will be available upon reasonable request to the corresponding author". All PLOS journals now require all data underlying the findings described in their manuscript to be freely available to other researchers, either 1. In a public repository, 2. Within the manuscript itself, or 3. Uploaded as supplementary information.

Additional Editor Comments (if provided):

Thank you for submission of your paper "A Pilot Acceptability Evaluation of MomMind: A Digital Health Intervention for Peripartum Depression Prevention and Management Focused on Health Disparities." As I understand it, the goal of the study was to evaluate the acceptability of a digital health tool called MomMind among low-income and minority women for managing peripartum depression. This paper meets all the requirements for publication. Additionally, all reviewers concur minor revisions and I agree with this. We look forward to your resubmission after these revisions have been made.

Reviewers' comments:

Reviewer's Responses to Questions

**Comments to the Author**

1. Does this manuscript meet PLOS Digital Health’s publication criteria? Is the manuscript technically sound, and do the data support the conclusions? The manuscript must describe methodologically and ethically rigorous research with conclusions that are appropriately drawn based on the data presented.

Reviewer #1: Yes

Reviewer #2: Yes

Reviewer #3: Yes

2. Has the statistical analysis been performed appropriately and rigorously?

Reviewer #1: Yes

Reviewer #2: Yes

Reviewer #3: Yes

3. Have the authors made all data underlying the findings in their manuscript fully available (please refer to the Data Availability Statement at the start of the manuscript PDF file)?

Reviewer #1: Yes

Reviewer #2: No

Reviewer #3: No

4. Is the manuscript presented in an intelligible fashion and written in standard English?

Reviewer #1: Yes

Reviewer #2: Yes

Reviewer #3: Yes

5. Review Comments to the Author

Reviewer #1: Thank you for the opportunity to review this paper. This is an interesting study and a nice pilot outcomes paper. My suggestions are as follows;

Please consider shortening the intro from pages 3-6, where it is essentially describing the previously published paper. The descriptions about the software and other detailed information is not needed here and is replicating previous work. The key elements of the intervention can be presented in a table, briefly summarised referencing previous papers, and then clearly stating what the gap this paper is filling. The last sentences of the introduction need to include the gap in the literature and the aim of this study and expected outcomes.

methods: please justify the sample size, for the quantitative outcomes and the qualitative outcomes.

The section under the heading Evaluation Procedures can be collapsed into a single table

Please present a figure showing the steps of the study, the study procedure, and then revisit the wording to see what can be stripped back as the method is very long.

Demographic data can be presented in a table (page 12), again improving readability.

The whole section on the interview themes is also well suited to a table, improving readability. (pages 14-16).

Please start the discussion with the aim of the study

The point that: Future digital health interventions targeting vulnerable populations should consider the use of specialized digital health frameworks addressing social determinants of health such as health literacy (an example being the extended Digilego framework we have presented in this study) for intervention design and development - is excellent. Can you refer to literature where this has been done? Can you expand on the consequences of not doing this?

Reviewer #2: Manuscript comments

- Including a table of participant characteristics would be very helpful, rather than reporting everything in the text. It would make the information easier to process. 

- How do you know your sample is representative of low-income women? Your target population is low-income perinatal women but your sample includes women who are not low-income.

- If the purpose of MomMind is to eventually be launched and used by the target population, will the digital features of MomMind only be website-based or will they also be available on a smartphone as an app? It would be good to describe the future plans for MomMind, especially since the target population may not have easy access to a computer or smartphone. If the purpose of the study was to only show the acceptability/utility of such a tool then this should be stated. 

- Were all distributions skewed? It might be nice to know this as justification for using the Wilcoxon test.

Text editing comments

- Need to cite Digilego in first sentence of “MomMind: A Health-Disparities Focused Digital Health Solution for PPD Self-Management” section

- In first sentence in Methods section, remove the word “has” from this line: Our mixed-methods study has involved the following steps:

- Typo in last sentence of Individual Interviews section: “intervies” should be “interviews”

- Same section, b) Instrumental Attitude: “and” should be “an”

- Same section, e) remove period at the end (after the question mark)

- In Peripartum Depression Health Literacy section, sentence “Average score on knowledge of PPD risk factors and causes at pre-intervention was 4.23, and at post-intervention 4.27, a change that is statistically unsignificant.” “unsignificant” should be “insignificant”

Reviewer #3: My Understanding of the Manuscript’s Intent:

The goal of the study was to evaluate the acceptability of a digital health tool called MomMind among low-income and minority women for managing peripartum depression. This pilot study aims to demonstrate the promising acceptability of a tailored digital health intervention for peripartum depression in underserved populations. The authors suggest enhancing design approaches for health equity.

My Review:

Given the interesting topic, solid evidence, and clear writing style, I would recommend the authors proceed with a submission to PLOS Digital Health after revising the data availability information and statistical analysis sections.

Originality - The study presents novel research evaluating a tailored digital health tool for an underserved population. The integration of behavior change and health literacy frameworks is an original approach.

Importance & Interest - Peripartum depression interventions for vulnerable groups are an important topic with broad appeal to digital health researchers and clinicians.

Rigor - The mixed methods evaluation uses appropriate surveys, interviews, and analyses. The sample size is decent for a pilot study but ultimately is small for power/generalizability.

Evidence - Results provide evidence that the tailored MomMind tool had high acceptability and improved women's depression knowledge. Qualitative data further supports the tool's benefits.

Utility & Accessibility - The tool aims to promote accessibility of peripartum depression resources for underserved women. The open-access publication benefits the broader community.

Open Science - The study methods and results are reported in enough detail to be reproduced. Data availability statements indicate data can be accessed upon request. The study involves human subjects research, which can justify data access restrictions to protect privacy. Overall, the authors have not yet fully met the data availability requirements due to lack of details and limiting access to just the corresponding author.

A Note on Statistical Analysis

The analysis in this study appears to have been carried out appropriately and with sufficient rigor. The analytic methods align with the study design and data characteristics. The analyses appear to have been conducted appropriately without obvious deficiencies. The statistical rigor is sufficient for the aims and scale of this pilot study. The reporting of methods and results is reasonably comprehensive and transparent. The analysis methods align with the study aims and data types. Using descriptive statistics, Wilcoxon signed-rank tests, and qualitative coding is appropriate, especially for a sample size of 30 participants since normal distribution cannot be assumed. The results section provides adequate numeric details from the analyses. Some key limitations around sample size and follow-up time are acknowledged. Changes in depression knowledge could be influenced by factors outside MomMind like prenatal classes. Controlling for confounders would strengthen conclusions. There is no control or comparison group, limiting the ability to attribute results solely to MomMind. The pre-post, non-randomized design cannot determine causality or directionality. Greater rigor would be experimental or RCT designs.

A Note on Data Access:

To properly comply, the authors should edit their data availability statement to:

- Provide an institutional/committee contact for data requests.

- Specify in detail the reasons and ethical approvals for restricted access.

- Clarify what specific data cannot be public and what could potentially be shared in an anonymous form.

An Additional Note on Equity:

The study focuses on evaluating a digital health tool specifically designed for an underserved population - low-income and minority women at risk for peripartum depression. This aligns well with the goals of promoting equity. The introduction highlights the disproportionate burden of peripartum depression in minority and low-SES groups as a motivation for the research.

- Within the remit of the US-based work, intersectionality could be better addressed - such as mental health access issues for low-income Hispanic women specifically, and an additional lens on potential barriers like digital literacy, disability, etc. that could affect. Incorporate perspectives from women facing multiple disparities. Discuss how UI design, literacy level, etc. were optimized for inclusivity.

- Nice to have for the future, the study sample is US-based, limiting (global) generalizability. Discuss how the digital health framework could be adapted for diverse cultural contexts. While not entirely within the remit of the pilot study, consider partnerships for evaluating the tool in low-resource global health settings. Low-income regions may have lower access to mobile devices needed to use the tool.

6. PLOS authors have the option to publish the peer review history of their article (<a href="https://journals.plos.org/digitalhealth/s/editorial-and-peer-review-process#loc-peer-review-history" target="_bl

---

## [Decision Letter · Decision Letter 1]

8 Apr 2024

A Pilot Acceptability Evaluation of MomMind: A Digital Health Intervention for Peripartum Depression Prevention and Management Focused on Health Disparities.

PDIG-D-23-00368R1

Dear Ms. Zingg,

We are pleased to inform you that your manuscript 'A Pilot Acceptability Evaluation of MomMind: A Digital Health Intervention for Peripartum Depression Prevention and Management Focused on Health Disparities.' has been provisionally accepted for publication in PLOS Digital Health.

Best regards,

Kara Burns

Guest Editor

PLOS Digital Health

Thank you for this submission. Congratulations on acceptance for publication.

Reviewer Comments (if any, and for reference):

Reviewer's Responses to Questions

**Comments to the Author**

1. If the authors have adequately addressed your comments raised in a previous round of review and you feel that this manuscript is now acceptable for publication, you may indicate that here to bypass the “Comments to the Author” section, enter your conflict of interest statement in the “Confidential to Editor” section, and submit your "Accept" recommendation.

Reviewer #1: All comments have been addressed

Reviewer #2: All comments have been addressed

2. Does this manuscript meet PLOS Digital Health’s publication criteria? Is the manuscript technically sound, and do the data support the conclusions? The manuscript must describe methodologically and ethically rigorous research with conclusions that are appropriately drawn based on the data presented.

Reviewer #1: Yes

Reviewer #2: Yes

3. Has the statistical analysis been performed appropriately and rigorously?

Reviewer #1: Yes

Reviewer #2: Yes

4. Have the authors made all data underlying the findings in their manuscript fully available (please refer to the Data Availability Statement at the start of the manuscript PDF file)?

Reviewer #1: Yes

Reviewer #2: Yes

5. Is the manuscript presented in an intelligible fashion and written in standard English?

Reviewer #1: Yes

Reviewer #2: Yes

6. Review Comments to the Author

Reviewer #1: Thank you for this revised manuscript it looks great!

Reviewer #2: The authors have addressed all Reviewer comments and the manuscript is much improved. I found one typo: <$200,000 should be >$200,000 in the household income table. Other than that the manuscript is ready for publication in my opinion.

7. PLOS authors have the option to publish the peer review history of their article (what does this mean?). If published, this will include your full peer review and any attached files.

**Do you want your identity to be public for this peer review?** For information about this choice, including consent withdrawal, please see our Privacy Policy.

Reviewer #1: No

Reviewer #2: **Yes: **Heather Mattie
